# Prebiotic Functions of Konjac Root Powder in Chocolate Milk Enriched with Free and Encapsulated Lactic Acid Bacteria

**DOI:** 10.3390/microorganisms10122433

**Published:** 2022-12-08

**Authors:** Said Ajlouni, Md. Nur Hossain, Ziqian Tang

**Affiliations:** 1School of Agriculture and Food, Faculty of Veterinary and Agricultural Sciences, The University of Melbourne, Melbourne, VIC 3010, Australia; 2Institute of Food Science and Technology, Bangladesh Council of Scientific and Industrial Research, Dhaka 1205, Bangladesh

**Keywords:** chocolate milk, konjac root powder, prebiotic, probiotics

## Abstract

This study investigated the prebiotic functions of Konjac root powder (KRP) when added to chocolate milk (ChM) enriched with 2% of free or microencapsulated lactic acid bacteria (FLAB or ELAB). The effects of different concentrations of KRP (0%, 2% and 4%) and refrigerated storage time on the physical, chemical and microbiological characteristics of this chocolate milk were examined. The results show that pH significantly declined (*p* < 0.05), while titratable acidity increased in all ChM samples with KRP and FLAB or ELAB during refrigerated storage. The pH values ranged from 6.0 ± 0.03 in samples enriched ELAB and 4% KRP to 6.33 ± 0.03 in ChM enriched with FLAB and 2% KRP. Viscosity of ChM was affected mainly by the added amounts of KRP and storage time. The largest viscosity (5500 cP) was observed in all samples containing 4% KPR on day zero and decreased significantly (*p* < 0.05) over storage time to reach 2800 cP in ChM samples containing 0% LAB and 4% KRP after 21 days of storage. Changes in LAB counts proved the initial hypothesis that KRP could act as prebiotics in the presence of LAB using chocolate milk as a carrier. The initial LAB counts in inoculated samples on day zero of refrigeration storage were not significantly different (*p* > 0.05) among all treatments. However, ChM enriched with 2% and 4% KRP and ELAB revealed significantly (*p* < 0.05) larger LAB counts (4.91 ± 0.78 and 5.0 ± 0.57 log CFU/mL, respectively) than the control (3.85 ± 0.55 log CFU/mL) after 21 days of storage.

## 1. Introduction

The paradigm of healthy lifestyle today has changed significantly, and people continue to increase their demands for more nutritious foods that can not only meet their basic daily needs, but with therapeutic functions [1]. Most recent investigations have been focusing on the utilization of probiotics as the living microorganism in food supplement that can provide beneficial effects to the host via balancing and improving the gut microflora [2]. Probiotics are defined as the living microorganism with healthy functions after consumption and digestion contributing to the improved host wellness [3]. The functions of probiotics in the human body involve, stimulating the digestibility, reducing the risk of some chronic disease such as diabetes, securing and enhancing the immune functions, and protecting human from pathogenic organisms [4,5]. Furthermore, it has been reported that probiotics can enhance the IgA secretion and securing the adaptive immunity. For instance, Karamese et al. [6] evaluated the immune function of a mixture of *Lactobacillus* and *Bifidobacterium* and concluded that the addition of probiotics can result in a remarkable increase in IgA concentration. Hence, foods fortified with probiotics can be considered among functional foods. Similarly, Rosa et al. reported in 2021 [7] that enriching dairy foods with prebiotics provide significant positive effect on product quality and consumers’ health. The same study illustrated that adding prebiotics to dairy foods improved their physical, chemical and microbial quality characteristics, and enhanced the consumers’ health through improvement in blood lipid profile and providing anti-diabetic and anti-hypertensive benefits to the consumers.

The concept of functional foods has received great attention, and functional food increasingly becomes an important factor that can influence consumers’ purchasing behaviour [8]. Functional foods refer to a food that is consumed as a part of daily diet, retains its conventional quality including appearance, flavour, and nutritional value, and at the same time increases the healthiness of consumers [9,10]. Functional dairy products have played a dominant role within the functional food segment, occupying about 40% of this market [1]. Among the functional dairy beverage, milk is highly appreciated and consumed worldwide due to its high-quality nutrient contents such as protein, minerals and fatty acids [11,12]. The combination of milk and probiotics via fermentation or directly utilizing milk as the probiotic carrier has been extensively studied and commercialized. Furthermore, chocolate milk, as the milk-related products with considerable popularity among consumers, regardless of their age, has been proved to be an ideal carrier for probiotics [13]. Consequently, using chocolate milk as a probiotic carrier in this current study was an ideal approach to investigate the role of konjact root powder (KRP) as a prebiotic source.

The prebiotic term was initially defined by Gibson and Roberfroid in 1995 as “non-digestible food ingredient that beneficially affects the host by selectively stimulating the growth and/or activity of one or a limited number of bacteria already resident in the colon”. However, the International Scientific Association for Probiotics and Prebiotics (ISAPP) modified that definition in 2017 to “a substrate that is selectively utilized by host microorganisms conferring a health benefit”. This modified definition considered non-carbohydrates substances such as phytochemicals as prebiotics, retained the need of selective microbiota, and proposed that prebiotic effect is not limited to the human gut, and may include animals [14]. The ISAPP panel also emphasised that prebiotics should be limited to substances that are metabolised by health promoting miroroganisms. Consequently, dietary fibres such as pectine, xylans and cellulose, which are metbolised by a broad range of gut microorganisms should not be classified as prebiotics [14]. The commonly used prebiotic compounds include froctooligosacharides, glucooligsacharides, inulin [14], D-allulose [15], β-glucan [16] and Indonesian porang tuber, which is rich in glucomannan [17].

Konjac root powder can be considered as a prebiotic functional food due to its high dietary fibre and glucomannan contents. Previous studies by Shang et al. [18]; Zeng et al. [19]; and Zhou et al. [20] mentioned that consuming Konjac root can decrease the accumulation of fat in the livers and reinforce the attenuation of oxidative stress in the body. These authors attributed such effects to the high contents of glucomannan in KRP. However, the applications of Konjac as a functional ingredient in the food system are still limited. Most of the Konjac usage has been focusing on its efficacy to encapsulate probiotics and to maintain their viability.

The prebiotic properties of KRP in dairy products, have not been fully studied and need further investigation. This study examined the possible prebiotics function of KRP when added at different concentrations to chocolate milk enriched with free and encapsulated probiotics.

## 2. Materials and Methods

### 2.1. Materials

Chocolate milk (Devondale Moo) was bought from a local market (Woolworth) in Melbourne, Vic., Australia. Microbial media (MRS agar), bacto peptone, and anaerobic sachets were bought from Thermo Fisher Scientific, Waltham, MA, USA (5 Caribbean Dive, Scoresby, Vic 3179, Australia). The lactic acid bacteria were kindly provided by Chr. Hansen, Bayswater, VIC, Australia. The Konjac powder was bought from ONSON KONJAC “https://www.google.com/search?client=firefox-b-e&q=ONSON+KONJAC (accessed on 18 March 2019)”. Other chemicals including phenolphthalein and NaOH were available in the food chemistry and microbiology laboratories in the school of agriculture and food at the University of Melbourne.

### 2.2. Methods

#### 2.2.1. Culture Activation

The frozen lactic acid bacteria included *Lactobacillus acidophilus* (La5) and *Bifidobacterium animalis* (Bb12) and were activated separately by mixing in pasteurized milk at room temperature, followed by incubation at 37 ℃ until the pH reached 5.8 (about 10 h). The activated cultures showed a microbial count, of 5 × 10^6^ to 5 × 10^7^ CFU/mL. The individually activated cultures were mixed at 1:1 ratio immediately after activation and used as a mixed culture. Half the activated mixed culture was left free, and the other half was encapsulated.

#### 2.2.2. Encapsulation

Whey protein solution (WPS) was prepared at 10% (*w*/*v*), using Milli-Q water and left overnight in the refrigerator at 4 °C. The WPS was pasteurised at 65 °C for 30 min in water bath before usage. The mixed probiotic cultures (50 mL, stock culture) was suspended in 1000 mL of 10% *w*/*v* whey protein solution and left for 2 h at room temperature to allow the interaction between whey protein and probiotic. The suspended probiotics were distributed into 50 mL Falcon tubes (~25 mL each) and frozen at −20 °C overnight before freeze-drying. These frozen samples were then freeze-dried at −50 °C for 72 h using benchtop freeze dryer (Dynavac FD3 Engineering, Melbourne, Australia The encapsulated LAB powder (ELAB) was stored at −20 °C until used [21]. The colony counts in the ELAB was determined by mixing 1 g ELAB with 50 mL pasteurized milk and left for 2 h at room temperature, before determining counts of the encapsulated LAB using serial dilutions and spread plate techniques [22]. The encapsulated cultures showed a microbial count of 6 $\times 10^{5}$ CFU/g EPP.

#### 2.2.3. Preparation of Chocolate Milk Samples

Commercially pasteurised chocolate milk (200 mL each) was aseptically distributed into 250 mL clean food quality glass container. Each chocolate milk sample (200 mL) was mixed with 4 mL of the activated LAB bacteria mixture (2% inoculum of free or encapsulated) and left for 2 h at room temperature to allow for the adjustment of the LAB in the chocolate milk. The inoculated chocolate milk samples were then fortified with 0%, 2%, 4% of Konjac root powder (KRP) in triplicate, mixed well and left for another 2 h at room temperature to allow for sufficient acclimatization. The control samples were prepared using chocolate milk fortified with 0%, 2%, 4% of KRP only (no added probiotics). All samples were stored refrigerated, and analyses were performed on day 0, 7, 14 and 21.

#### 2.2.4. Analyses of Physio-Chemical and Microbial Characteristics of the Prepared Chocolate Milk Samples

##### pH

The pH was measured using Dedicated pH/ORP meter-HI 2002 (HANNA instruments, Woonsocket, RI, USA). The pH meter was calibrated using two standard buffer solutions (pH 4 and 8). The pH electrodes were rensid, plot dried and immersed in 10 mL chocolate milk (ChM) samples. The index of pH was obtained directly following the machine operating manual.

##### Viscosity

The viscosity was measured by the DV1 Digital viscometer (Brookfieled, Toronto, ON, Canada). ChM samples were wormed to room temperature and 50 mL from each was placed in a small beaker before inserting the proper spindle. The index of viscosity was obtained directly following the Brookfield Dial Viscometer operating manual [23]. The results are reported in cP.

##### Titratable Acidity

Titratable acidity (TA) was performed following the method of Chuah et al. [24]. Each tested sample (10 mL) was mixed with 90 mL Milli-Q water in a conical flask, mixed with 3–4 drops of phenolphthalein and titrated using 0.1 mol/L NaOH solution. The TA was calculated as percentage lactic acid content and calculated using the following equation:$$\mathrm{TA}=(\mathrm{Milliliters}\mathrm{of}\mathrm{NaOH}\times 10^{-3}\times 0.1\times 90.08\times 100)÷10$$

##### LAB Bacteria Count

The LAB bacteria count was performed using a spot plate technique following the method of James [25] with slight modification. On each day of analysis, the samples were serially diluted up to $10^{-4}\;$ using 0.1% sterile bacto peptone. The spot plate count involved micro pipetting 10 µL of each selected dilution into each of 10 spots already marked on the plate. This technique does not require spreading. The inoculated plates were incubated anaerobically through placing the plates upside down in the anaerobic chamber with anaerobic gas generation at 37 °C for 24 h.

##### Statistical Analysis

The design of this study involved three trials, and the measurements of each tested parameter within each trial were repeated at least 3 times. The data were statistically analysed using a Minitab^®^19 statistical software, 2019 and one way ANOVA and the means were separated based on LSD at 95% confidence level. The results are presented as the mean ± standard deviation (*n* = 9).

## 3. Results

### 3.1. Effect of Added KRP on the Physio-Chemical Properties Chocolate Milk

#### 3.1.1. pH

The results of pH measurements of individual treatments during refrigerated storage of ChM are shown in Figure 1. The recoded pH values indicated a gradual and significant (*p* < 0.05) decrease in all samples fortified with KRP and free LAB (FLAB) and encapsulated LAB (ELAB) during refrigerated storage. The final pH values in these ChM samples after 21 days of refrigerated storage ranged from the smallest pH of 6.05 ± 0.03 in samples enriched with 4% KRP and ELAB to the largest pH (6.33 ± 0.03) in ChM enriched with 2% KRP and FLAB. The control samples that were not enriched with KRP, or probiotics also showed significant decline (*p* < 0.05) in pH after day 21 of storage. The pH values in these control samples declined from 6.67 ± 0.03 on day zero to 6.65 ± 0.01 on day 21. Furthermore, samples enriched with 2% and 4% KRP only (no added probiotics), revealed similar gradual and significant decline (*p* < 0.05) in pH values during storage (Figure 1).

#### 3.1.2. Titratable Acidity

The titratable acidity (TA) and pH are always closely and negatively related to each other, which means the higher the pH in a given food sample, the lower the titratable acidity and vice versa. Following the opposite pattern with the detected decreases in pH values, data in Table 1. showed significant increment (*p* < 0.05) in the %TA even in the control (no added LAB or KRP) ChM samples after 21 days of refrigerated storage. The %TA reported as lactic acid content in the control samples was 0.18 ± 0.0 % after 21 days of storage. Furthermore, enriching ChM with LAB in the presence or absence of KRP caused significant increases (*p* < 0.05) in the %TA in all inoculated samples. The %TA in ChM enriched with 2% FLAB (no added KRP) increased significantly (*p* < 0.05) from 0.17 ± 0.0% on day zero to 0.20 ± 0.0% on day 21 of refrigerated storage. Similar trends were also detected in ChM samples enriched with 2% LAB (free or encapsulated) and 2% or 4% KRP. Adding 2% of FLAB in the presence of 2% or 4% KRP increased the %TA from 0.18 ± 0.0% to 0.23 ± 0.01 and 0.24 ± 0.0%, respectively after 21 days of refrigerated storage (Table 1).

#### 3.1.3. Viscosity

There was a positive correlation between the amounts of added KRP in ChM and viscosity. The data in Figure 2 clearly illustrate that regardless of the addition of LAB or not, ChM samples with 4% KRP had significantly (*p* < 0.05) greater viscosity than samples enriched with 2% KRP during the whole storage period. The same data also revealed that viscosity was below the detection level (100 cP) in the absence of KRP. Viscosity of ChM containing 0% LAB and 2 % KRP on day zero was 1400 cP, and significantly increased (*p* < 0.05) to 5500 cP when the amount of added KRP increased from 2% to 4%. Similar viscosity was recorded in all samples containing 4% KRP and was not affected by the presence of absence of free or encapsulated probiotics.

### 3.2. Effect of Added KRP on LAB Counts in Chocolate Milk

The data in Table 2 represent changes in the LAB (probiotic) counts (log CFU/mL) in chocolate milk (ChM) during refrigerated storage. Treatments included the control (C), where chocolate milk was enriched with KRP only (no added LAB) and chocolate milk inoculated with 2% of free LAB (FLAB) or encapsulated LAB (ELAB) and enriched with 0%, 2% and 4% of Konjac root powder (KRP). The LAB counts in the control samples (C) were <100 CFU/mL (the minimum detection limits). The same data in Table 2 show that enriching ChM with KRP supported the survival of added LAB during refrigerated storage. Although the initial LAB counts on day zero of refrigeration storage were not significantly different (*p* > 0.05) among all samples enriched with 2% LAB and various concentrations of KRP, ChM enriched with 2% and 4% KRP revealed significantly (*p* < 0.5) larger LAB counts than the samples containing (0% KRP) after 21 days of storage. The final LAB counts in ChM containing 4% KRP in the presence of both FLAB and ELAB after 21 days of refrigerated storage were 4.91 ± 0.28 and 5.0 ± 0.57 log CFU/mL, respectively. These log counts were significantly larger (*p* < 0.05) than those in ChM samples with no added KRP which showed 4.73 ± 0.60 log CFU/mL and 3.85 ± 0.55 log CFU/mL in FLAB and ELAB, respectively. Similar trends were also recorded with all treatments after 14 days of refrigerated storage, while no significant differences (*p* > 0.05) in LAB counts between treatments were detected at 0 and 7 days of storage.

## 4. Discussion

### 4.1. Effect of Added KRP on the Physio-Chemical Properties Chocolate Milk

#### 4.1.1. pH

The pH results indicate that adding free or encapsulated probiotics and KRP to chocolate milk contributed to the decline in pH values during storage. These observations agreed with previous studies by Daneshi et al. [26] and Valencia et al. [27] who mentioned that during the storage of the milk-based products containing LAB, the pH decreased, and the acidity increased reversely. Another study by Holzapfel and Schillinger [28] reported decreases in pH during storage of milk-based product and attributed that to the activity of LAB. Similarly, these authors confirmed that LAB in milk-based product would continue their growth and metabolism during storage causing a decline in pH. It should be explained here that the significant decline (*p* < 0.05) in the pH values in ChM samples which did not have added probiotics or KRP could be attributed to the presence of some residual LAB in the chocolate milk samples.

The decline in pH values reported in this current study were much smaller than those reported by Daneshi et al. [26]. These authors tested the viability of the several strains of LAB in the milk-carrot juice mixtures. They reported that after 20 days of storage at 4 °C the pH of the milk/carrot juice mixtures containing *L. rhamnosus* drops from 6.47 to 5.33. Such a significant decrease in pH might be caused by the presence of carrot juice in that case. Although, enriching ChM with KRP and LAB caused significant decline (*p* < 0.05) in pH after the 21 days of refrigerated storage, the final pH values were larger than 6.1. Consequently, the ChM is not acidic at all, and such pH values (6.1) should not affect the consumer acceptance.

#### 4.1.2. Titratable Acidity

The observed changes in %TA of ChM enriched with KRP agreed with those reported by Valencia et al. [27] who studied creamy milk chocolate enriched with LAB and fructo-oligosaccharides. They noted significant decrease in pH from 6.1 to 5.1 and increases in % TA from 0.28% to 0.6% after the 28 days of refrigerated storage. Another study by Holzapfel and Schillinger [28] explained the decrease in pH and increase in %TA of milk-based products to the activity of LAB during transportation and storage. LAB in milk-based products will continue their metabolic activities and convert lactose into lactic acid, subsequently result in decreasing the pH and increasing the acidity of the food. However, microbial growth and their metabolic activities will be affected by various factors, including storage temperature. Refrigeration usually causes significant decline in bacterial growth and metabolic activities. The results from this current study and other previously reported investigations [27,28] confirmed this observation via reporting minimal changes in pH and TA during refrigerated storage. The lowest pH of ChM after 21 days of storage was close to 6.1 and the highest %TA was 0.36%, meaning that the ChM was not acidic even after 21 days of storage. Consequently, it was concluded that the suggested prebiotic function of KRP was not strong enough under such refrigeration conditions due to the slow LAB metabolic activities.

The same results also reveal that increments in the %TA were greater using encapsulated LAB as opposed to free LAB. The %TA reached 0.24 ± 0.0% and 0.36 ± 0.01 in the presence of FLAB and ELAB and 4% KRP, respectively. These observations suggested that encapsulating LAB could provide better protection and survivability to the LAB, which ended up producing more lactic acid than the FLAB. These findings agreed with those of Rajaie et al. [29] how examined the effect of encapsulation on the survival of *Enterococcus faecium* in dairy probiotic beverages. The same authors reported that pH depletion and acidity increases were more dramatic in the presence of ELAB than the FLAB in cultured beverages.

#### 4.1.3. Viscosity

Changes in the viscosity of ChM fortified with KRP were supported by the finding of Yoshimura and Nishinari [30] who examined the viscosity of gelation formed by KGM and concluded that the higher the concentration of KGM in water, the higher the viscosity. A similar conclusion was also reached by Rong et al. [31] who reported that the increased concentration of KOS (KGM hydrolysate) in yogurt will increase its viscosity significantly. Furthermore, Tobin et al. [32] also recorded a noticeable increase in the viscosity of skim milk with increased konjac concentration. The same authors suggested that increasing viscosity of milk in the presence of KGM, is caused by the formation of a firm gel system through the interaction between protein, water, milk salt and the KGM polysaccharide. It is anticipated that KGM as a biopolymer can decrease the force between water droplets, inducing the flocculation of droplets and increasing the viscosity of the emulsion.

The current research results also demonstrate that viscosity was affected by storage time and declined gradually and significantly (*p* < 0.05) in all treatments during refrigerated storage. The largest viscosity (5500 cP) was observed in all samples containing 4% KPR on day zero and decreased significantly (*p* < 0.05) over storage time to reach about 2800 cP in ChM samples with 0 LAB and 4% KRP after 21 days of storage. Results in Figure 2 also showed that the most significant decline in viscosity was between days 14 and 21 in ChM samples containing 0% LAB and 4% KRP. Such a significant decline in viscosity could be attributed to the partial hydrolysis of KRP by LAB and decrease in pH values during that extended period of storage. Other studies by Chenlo et al. [33] and Rong et al. [31] attributed the decline in viscosity during the refrigerated storage to the hydrolysis reactions and degradation of the gel-network formed by the polysaccharide-protein interaction between KGM and milk-based product.

### 4.2. Effect of Added KRP on LAB Counts in Chocolate Milk

The detected changes in the LAB (probiotic) counts (log CFU/mL) in chocolate milk (ChM) enriched with KRP during refrigerated storage confirmed the positive effect of KRP in supporting the survival of LAB in ChM during long (>14 days) refrigerated storage (Table 2). The same data also reveal that using 4% KRP had significantly better effects (*p* < 0.05) on the survival of FLAB than 2% KRP after 14 and 21 days of storage. However, such differences between 2% and 4% KRP on the survival of LAB were not detected when using encapsulated LAB. The greater effect of 4% than 2% KRP in the presence of FLAB may be attributed to the better prebiotic function of KRP in the presence of free LAB. However, enriching ChM with KRP prevented any significant decline in the ELAB counts during refrigerated storage. The decline in the ELAB count in ChM samples enriched with 4% KRP was only 0.17 log CFU/mL as compared with 1.34 log CFU/mL in the absence of KRP (0%) after 21 day of storage (Table 2). The prebiotic function of Konjac oligosaccharides (KOS) and fibre has been reported in fermented dairy products. Rong et al. [31] indicated that enriching yoghurt with 0.8% KOS yielded significantly higher LAB counts than the control after 14 days of storage. However, in the current study, the whole KRP was added rather than the KOS.

The positive effect of KRP on the growth and survival of LAB in ChM can be attributed to the high contents of soluble fibre in KRP [34], which provided the prebiotic function. A study by Al-Ghazzewi et al. [35] reported that high fibre content in KRP could be the major carbon sources for LAB. Furthermore, the addition of KRP was reported to strengthen the gel structure in the milk and milk-based products such as yogurt and cheese. This gelling function could be attributed to the large number of hydroxyl groups in KRP and the formation of hydrogen bonds with the amino groups of milk proteins [31,35]. Such gel formation in milk could increase its buffering capacity, resulting in a good survivability of LAB in ChM during refrigerated storage. These recorded positive effects of added KRP and its function as a prebiotic in chocolate milk, will help the food industry to utilize konjac root powder as a source of fibre and glucomannan. It is anticipated that KRP can be applied to non-dairy foods, such as different types of soup and fruit juices to improve their functional properties and contribute to better human health.

## 5. Conclusions

The addition of KRP can support the growth and survival of LAB in chocolate milk. With the increasing concentration of KRP in chocolate milk, the good survivability rate of the LAB in chocolate milk was maintained after 21 days of storage at 4 °C. The protection of LAB, the decline in pH, increase in titratable acidity and viscosity of ChM were clearly documents and attributed to the added KRP. Moreover, the encapsulation of LAB could provide better preservation and survivability of the LAB in chocolate milk during refrigerated storage. The results also show that prolonged storage time caused significant in viscosity. The prebiotic function of Konjac root powder enables it to be applied into both dairy and non-dairy foods. KRP will not only improves food quality, but also contributes to a better human health upon consumption of food enriched with KRP.

This study explored the prebiotic potential of KRP in a new type of probiotic food-LAB cultured chocolate milk and developed a new type of food combined by the KRP and chocolate milk with health functions. The demonstrated prebiotic functional properties of KRP in chocolate milk, will help the food industry utilizing konjac root powder as a source of fibre and glucomannan. It is anticipated that KRP can also be applied to non-dairy foods, such as different types of soups, snacks and fruit juices to improve their functional properties and contribute to better human health.

However, it is recommended that further studies are needed to explore these positive effects of KRP in the presence of different species of probiotics. Finally, including sensory tastings in the future studies will be essential to judge the consumer acceptance of such food products.

## Figures and Tables

**Figure 1 microorganisms-10-02433-f001:**
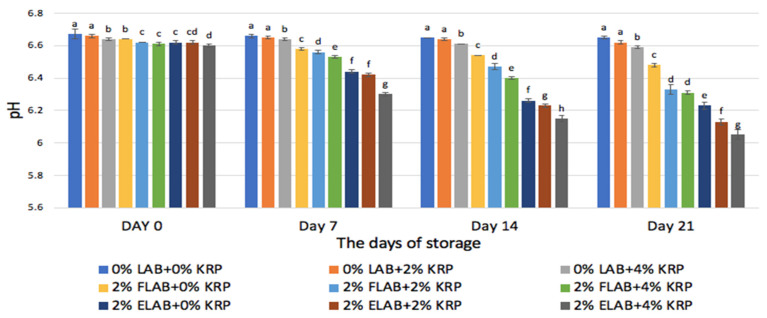
The impact of storage time on the pH of chocolate milk enriched with 0% or 2% LAB and various concentration (0%, 2%, and 4%) of KRP. Columns within each day of measurement with different superscript letters are significantly different (95% confidence level).

**Figure 2 microorganisms-10-02433-f002:**
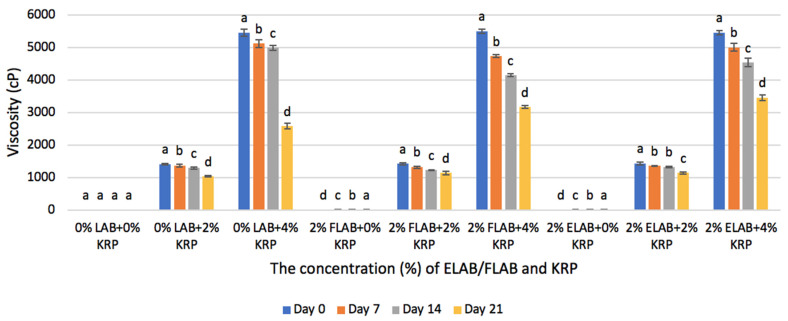
The impact of various concentration of KRP on the Viscosity (cP) of chocolate milk that contains 0% LAB, 2% ELAB and 2% FLAB, during the 0, 7, 14 and 21 days of storage. Standard deviation (SD) is presented in the forms of error bars. Columns with different superscript letters are significantly different (95% confidence level) within each group).

**Table 1 microorganisms-10-02433-t001:** The impact of storage time on the titratable acidity (% lactic acid content) of chocolate milk enriched with 0% or 2% (FLAB or ELAB) and different concentrations (0%, 2% and 4%) of KRP. Data are presented as mean ± standard deviation. Means in a row followed with different superscript letters are significantly different (95% confidence level).

% of Added LAB	Type of Added LAB	Amounts of Added KRP (%)	Storage Time (Days)			
			0	7	14	21
0	N/A	0	$0.17\pm {0.01\;}^{a}$	$0.18\pm 0.00^{\;a}$	$0.18\pm 0.00^{\;a}$	$0.18\pm 0.00{\;}^{b}$
		2	$0.17\pm {0.00\;}^{c}$	$0.18\pm {0.00\;}^{b}$	$0.18\pm {0.01\;}^{b}$	$0.18\pm {0.00\;}^{a}$
		4	$0.17\pm {0.00\;}^{c}$	$0.18\pm 0.00{\;}^{b}$	$0.18\pm 0.00^{\;b}$	$0.18\pm 0.00^{\;a}$
2	FLAB	0	$0.17\pm {0.00\;}^{d}$	$0.18\pm 0.00^{\;c}$	$0.20\pm 0.01^{\;b}$	$0.20\pm 0.00{\;}^{a}$
		2	$0.18\pm {0.00\;}^{c}$	$0.18\pm 0.00{\;}^{c}$	$0.21\pm 0.00{\;}^{b}$	$0.23\pm 0.01^{\;a}$
		4	$0.18\pm {0.00\;}^{d}$	$0.19\pm 0.01{\;}^{c}$	$0.22\pm 0.00{\;}^{b}$	$0.24\pm 0.00{\;}^{a}$
	ELAB	0	$0.18\pm {0.00\;}^{d}$	$0.21\pm {0.00\;}^{c}$	$0.27\pm 0.01^{\;b}$	$0.29\pm 0.00{\;}^{a}$
		2	$0.18\pm {0.00\;}^{d}$	$0.21\pm 0.00^{\;c}$	$0.29\pm 0.01^{\;b}$	$0.32\pm 0.00{\;}^{a}$
		4	$0.18\pm {0.00\;}^{d}$	$0.24\pm 0.00^{\;c}$	$0.32\pm 0.00^{\;b}$	$0.36\pm 0.01{\;}^{a}$

**Table 2 microorganisms-10-02433-t002:** Impact of various concentrations of Konjac root powder (KRP) and storage time on the counts (log CFU/mL) of lactic acid bacteria (LAB) in chocolate milk inoculated with free LAB (FLAB) or encapsulated (ELAB). Data are presented as mean ±  SD (*n* = 9). Means followed with a different small superscript letter in a raw and capital superscript in a column are significantly different (*p* < 0.05).

Amounts of Added LAB (%)	Type of Added LAB	Amounts of Added KRP (%)	Storage Time (Days)			
			0	7	14	21
0	None (Control)	0	<100	<100	<100	<100
		2	<100	<100	<100	<100
		4	<100	<100	<100	<100
2	FLAB	0	5.20$\;\pm \;0.77{\;}^{a}$	5.12$\;\pm \;0.64^{\;b}$	5.01$\;\pm \;0.41{\;}^{Ac}$	4.73$\;\pm \;0.60{\;}^{Ad}$
		2	5.19$\;\pm \;0.57^{\;a}$	5.11$\;\pm \;0.60{\;}^{b}$	5.05$\;\pm \;0.50{\;}^{Ac}$	4.82$\;\pm \;0.55{\;}^{Bd}$
		4	5.19$\;\pm \;0.97{\;}^{a}$	5.15$\;\pm \;0.43^{\;a}$	5.11$\;\pm \;0.45{\;}^{Bab}$	4.91$\;\pm \;0.28{\;}^{Cd}$
	ELAB	0	5.19$\;\pm \;0.69{\;}^{a}$	5.05$\;\pm \;0.38{\;}^{b}$	4.97$\;\pm \;0.45{\;}^{Cc}$	3.85$\;\pm \;0.55{\;}^{Dd}$
		2	5.18$\;\pm \;0.62{\;}^{a}$	5.09$\;\pm \;0.36^{\;b}$	5.02$\;\pm \;0.49{\;}^{ACc}$	4.91$\;\pm \;0.78{\;}^{Cd}$
		4	5.17$\;\pm \;0.51{\;}^{a}$	5.09$\;\pm \;0.43{\;}^{b}$	5.06$\;\pm \;0.55{\;}^{ABb}$	$5.0\pm 0.57{\;}^{Ebc}$

## Data Availability

Not applicable.

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
