# Peer review of "Prebiotic Functions of Konjac Root Powder in Chocolate Milk Enriched with Free and Encapsulated Lactic Acid Bacteria"

_microorganisms, 2022, doi:10.3390/microorganisms10122433_

Round 1

Reviewer 1 Report

x recent prebiotic definition must be given (Gibson et al., 2017)

x the use of prebiotic in dairy foods must be emphasiized by recent references ( International Dairy JournalVolume 117, June 2021, 105009)

x studies about prebiotic ingredients must be added ( Food Science and Technology, 2022, Volume 42 elocation e06321 ;Food Science and Technology,  2022, Volume 42 elocation e10122; Food Science and Technology,  2022, Volume 42 elocation e07022)

x some analysis do not contain sample preparation. Please include these part.

x  why chocolate milk? Please justify and use recent studies about  this food (Int J dairy technology, Volume74Issue1, February 2021, Pages 246-257)

x more discussion is need and the relevance for the food industry is lacking

Reviewer 2 Report

The manuscript entitled: “Prebiotic functions of Konjac root powder in chocolate milk enriched with free and encapsulated lactic acid bacteria”  is a very interesting article that describes prebiotic functions of Konjac root powder when added to chocolate milk of free or microencapsulated lactic acid bacteria. The article has been written well, however, needs major revision. Here are some remarks that have to be considered in further submission.   

1. Figures 1-2 are not clear and should be made with better resolution and high quality, it is the same for tables.

2. I suggest to authors to prepare graphical abstract with  high-quality resolution

3. Conclusion section must be better described

4. English language spelling grammar of the manuscript should be checked 

5. Please check the references style

Round 2

Reviewer 2 Report

accept in present form